# Informative Features for Model Comparison

**Wittawat Jitkrittum**
Max Planck Institute for Intelligent Systems
wittawat@tuebingen.mpg.de

**Heishiro Kanagawa**
Gatsby Unit, UCL
heishirok@gatsby.ucl.ac.uk

**Patsorn Sangkloy**
Georgia Institute of Technology
patsorn_sangkloy@gatech.edu

**James Hays**
Georgia Institute of Technology
hays@gatech.edu

**Bernhard Schölkopf**
Max Planck Institute for Intelligent Systems
bernhard.schoelkopf@tuebingen.mpg.de

**Arthur Gretton**[*]
Gatsby Unit, UCL
arthur.gretton@gmail.com

## Abstract

Given two candidate models, and a set of target observations, we address the problem of measuring the relative goodness of fit of the two models. We propose two new statistical tests which are nonparametric, computationally efficient (runtime complexity is linear in the sample size), and interpretable. As a unique advantage, our tests can produce a set of examples (informative features) indicating the regions in the data domain where one model fits significantly better than the other. In a real-world problem of comparing GAN models, the test power of our new test matches that of the state-of-the-art test of relative goodness of fit, while being one order of magnitude faster.

## 1 Introduction

One of the most fruitful areas in recent machine learning research has been the development of effective generative models for very complex and high dimensional data. Chief among these have been the generative adversarial networks [Goodfellow et al., 2014, Arjovsky et al., 2017, Nowozin et al., 2016], where samples may be generated without an explicit generative model or likelihood function. A related thread has emerged in the statistics community with the advent of Approximate Bayesian Computation, where simulation-based models without closed-form likelihoods are widely applied in bioinformatics applications [see Lintusaari et al., 2017, for a review]. In these cases, we might have several competing models, and wish to evaluate which is the better fit for the data.

The problem of model criticism is traditionally defined as follows: how well does a model $Q$ fit a given sample $Z_n := \{z_i\}_{i=1}^n \overset{i.i.d.}{\sim} R$? This task can be addressed in two ways: by comparing samples $Y_n := \{y_i\}_{i=1}^n$ from the model $Q$ and data samples, or by directly evaluating the goodness of fit of the model itself. In both of these cases, the tests have a null hypothesis (that the model agrees with the data), which they will reject given sufficient evidence. Two-sample tests fall into the first category: there are numerous nonparametric tests which may be used [Alba Fernández et al., 2008, Gretton et al., 2012a, Friedman and Rafsky, 1979, Székely and Rizzo, 2004, Rosenbaum, 2005, Harchaoui et al., 2008, Hall and Tajvidi, 2002, Jitkrittum et al., 2016], and recent work in applying two-sample tests to the problem of model criticism [Lloyd and Ghahramani, 2015]. A second approach requires the model density $q$ explicitly. In the case of simple models for which normalisation is not an issue (e.g., checking for Gaussianity), several tests exist [Baringhaus and Henze, 1988, Székely and Rizzo,

---

[*] Arthur Gretton's ORCID ID: 0000-0003-3169-7624.

2005]; when a model density is known only up to a normalisation constant, tests of goodness of fit have been developed using a Stein-based divergence [Chwialkowski et al., 2016, Liu et al., 2016, Jitkrittum et al., 2017b].

An issue with the above notion of model criticism, particularly in the case of modern generative models, is that *any* hypothetical model $Q$ that we design is likely a poor fit to the data. Indeed, as noted in Yamada et al. [2018, Section 5.5], comparing samples from various Generative Adversarial Network (GAN) models [Goodfellow et al., 2014] to the reference sample $Z_n$ by a variant of the Maximum Mean Discrepancy (MMD) test [Gretton et al., 2012a] leads to the trivial conclusion that all models are wrong [Box, 1976], i.e., $H_0\colon Q = R$ is rejected by the test in all cases. A more relevant question in practice is thus: "Given two models $P$ and $Q$, which is closer to $R$, and in what ways?" This is the problem we tackle in this work.

To our knowledge, the only nonparametric statistical test of *relative* goodness of fit is the Rel-MMD test of Bounliphone et al. [2015], based on the maximum mean discrepancy [MMD, Gretton et al., 2012a]. While shown to be practical (e.g., for comparing network architectures of generative networks), two issues remain to be addressed. Firstly, its runtime complexity is quadratic in the sample size $n$, meaning that it can be applied only to problems of moderate size. Secondly and more importantly, it does not give an indication of where one model is better than the other. This is essential for model comparison: in practical settings, it is highly unlikely that one model will be uniformly better than another in all respects: for instance, in hand-written digit generation, one model might produce better "3"s, and the other better "6"s. The ability to produce a few examples which indicate regions (in the data domain) in which one model fits better than the other will be a valuable tool for model comparison. This type of interpretability is useful especially in learning generative models with GANs, where the "mode collapse" problem is widespread [Salimans et al., 2016, Srivastava et al., 2017]. The idea of generating such distinguishing examples (so called *test locations*) was explored in Jitkrittum et al. [2016, 2017b] in the context of model criticism and two-sample testing.

In this work, we propose two new linear-time tests for relative goodness-of-fit. In the first test, the two models $P, Q$ are represented by their two respective samples $X_n$ and $Y_n$, and the test generalises that of Jitkrittum et al. [2016]. In the second, the test has access to the probability density functions $p, q$ of the two respective candidate models $P, Q$ (which need only be known up to normalisation), and is a three-way analogue of the test of Jitkrittum et al. [2017b]. In both cases, the tests return locations indicating where one model outperforms the other. We emphasise that the practitioner must choose the model ordering, since as noted earlier, this will determine the locations that the test prioritises. We further note that the two tests complement each other, as both address different aspects of the model comparison problem. The first test simply finds the location where the better model produces mass closest to the test sample: a worse model can produce too much mass, or too little. The second test does not address the overall probability mass, but rather the shape of the model density: specifically, it penalises the model whose derivative log density differs most from the target (the interpretation is illustrated in our experiments). In the experiment on comparing two GAN models, we find that the performance of our new test matches that of Rel-MMD while being one order of magnitude faster. Further, unlike the popular Fréchet Inception Distance (FID) [Heusel et al., 2017] which can give a wrong conclusion when two GANs have equal goodness of fit, our proposed method has a well-calibrated threshold, allowing the user to flexibly control the false positive rate.

## 2    Measures of Goodness of Fit

In the proposed tests, we test the relative goodness of fit by comparing the relative magnitudes of two distances, following Bounliphone et al. [2015]. More specifically, let $D(P, R)$ be a discrepancy measure between $P$ and $R$. Then, the problem can be formulated as a hypothesis test proposing $H_0\colon D(P, R) \leq D(Q, R)$ against $H_1\colon D(P, R) > D(Q, R)$. This is the approach taken by Bounliphone et al. who use the MMD as $D$, resulting in the relative MMD test (Rel-MMD). The proposed Rel-UME and Rel-FSSD tests are based on two recently proposed discrepancy measures for $D$: the Unnormalized Mean Embeddings (UME) statistic [Chwialkowski et al., 2015, Jitkrittum et al., 2016], and the Finite-Set Stein Discrepancy (FSSD) [Jitkrittum et al., 2017b], for the sample-based and density-based settings, respectively. We first review UME and FSSD. We will extend these two measures to construct two new relative goodness-of-fit tests in Section 3. We assume throughout that the probability measures $P, Q, R$ have a common support $\mathcal{X} \subseteq \mathbb{R}^d$.

**The Unnormalized Mean Embeddings (UME) Statistic** UME is a (random) distance between two probability distributions [Chwialkowski et al., 2015] originally proposed for two-sample testing for

$H_0 : Q = R$ and $H_1 : Q \neq R$. Let $k_Y : \mathcal{X} \times \mathcal{X} \to \mathbb{R}$ be a positive definite kernel. Let $\mu_Q$ be the mean embedding of $Q$, and is defined such that $\mu_Q(\boldsymbol{w}) := \mathbb{E}_{\boldsymbol{y} \sim Q} k(\boldsymbol{y}, \boldsymbol{w})$ (assumed to exist) [Smola et al., 2007]. Gretton et al. [2012a] shows that when $k_Y$ is characteristic [Sriperumbudur et al., 2011], the Maximum Mean Discrepancy (MMD) *witness function* $\mathrm{wit}_{Q,R}(\boldsymbol{w}) := \mu_Q(\boldsymbol{w}) - \mu_R(\boldsymbol{w})$ is a zero function if and only if $Q = R$. Based on this fact, the UME statistic evaluates the squared witness function at $J_q$ test locations $W := \{\boldsymbol{w}_j\}_{j=1}^{J_q} \subset \mathcal{X}$ to determine whether it is zero. Formally, the population squared UME statistic is defined as $U^2(Q, R) := \frac{1}{J} \sum_{j=1}^{J} (\mu_Q(\boldsymbol{w}_j) - \mu_R(\boldsymbol{w}_j))^2$. For our purpose, it will be useful to rewrite the UME statistic as follows. Define the feature function $\psi_W(\boldsymbol{y}) := \frac{1}{\sqrt{J_q}} \big( k_Y(\boldsymbol{y}, \boldsymbol{w}_1), \dots, k_Y(\boldsymbol{y}, \boldsymbol{w}_{J_q}) \big)^\top \in \mathbb{R}^{J_q}$. Let $\psi_W^Q := \mathbb{E}_{\boldsymbol{y} \sim Q}[\psi_W(\boldsymbol{y})]$, and its empirical estimate $\hat{\psi}_W^Q := \frac{1}{n} \sum_{i=1}^{n} \psi_W(\boldsymbol{y}_i)$. The squared population UME statistic is equivalent to $U^2(Q, R) := \|\psi_W^Q - \psi_W^R\|_2^2$. For $W \sim \eta$ where $\eta$ is a distribution with a density, Theorem 2 in Chwialkowski et al. [2015] states that if $k_Y$ is real analytic, integrable, and characteristic, then $\eta$-almost surely $\|\psi_W^Q - \psi_W^R\|_2^2 = 0$ if and only if $Q = R$. In words, under the stated conditions, $U(Q, R) := U_Q$ defines a distance between $Q$ and $R$ (almost surely).[2] A consistent unbiased estimator is $\widehat{U_Q^2} = \frac{1}{n(n-1)} \big[\| \sum_{i=1}^{n} [\psi_W(\boldsymbol{y}_i) - \psi_W(\boldsymbol{z}_i)]\|^2 - \sum_{i=1}^{n} \|\psi_W(\boldsymbol{y}_i) - \psi_W(\boldsymbol{z}_i)\|^2 \big]$, which clearly can be computed in $\mathcal{O}(n)$ time. Jitkrittum et al. [2016] proposed optimizing the test locations $W$ and $k_Y$ so as to maximize the test power (i.e., the probability of rejecting $H_0$ when it is false) of the two-sample test with the normalized version of the UME statistic. It was shown that the optimized locations give an interpretable indication of where $Q$ and $R$ differ in the input domain $\mathcal{X}$.

**The Finite-Set Stein Discrepancy (FSSD)** FSSD is a discrepancy between two density functions $q$ and $r$. Let $\mathcal{X} \subseteq \mathbb{R}^d$ be a connected open set. Assume that $Q, R$ have probability density functions denoted by $q, r$ respectively. Given a positive definite kernel $k_Y$, the *Stein witness function* [Chwialkowski et al., 2016, Liu et al., 2016] $\boldsymbol{g}^{q,r} : \mathcal{X} \to \mathbb{R}^d$ between $q$ and $r$ is defined as $\boldsymbol{g}^{q,r}(\boldsymbol{w}) := \mathbb{E}_{\boldsymbol{z} \sim r}[\boldsymbol{\xi}^q(\boldsymbol{z}, \boldsymbol{w})] = (g_1^{q,r}(\boldsymbol{w}), \dots, g_d^{q,r}(\boldsymbol{w}))^\top$, where $\boldsymbol{\xi}^q(\boldsymbol{z}, \boldsymbol{w}) := k_Y(\boldsymbol{z}, \boldsymbol{w}) \nabla_{\boldsymbol{z}} \log q(\boldsymbol{z}) + \nabla_{\boldsymbol{z}} k_Y(\boldsymbol{z}, \boldsymbol{w})$. Under appropriate conditions (see Chwialkowski et al. [2016, Theorem 2.2] and Liu et al. [2016, Proposition 3.3]), it can be shown that $\boldsymbol{g}^{q,r} = \boldsymbol{0}$ (i.e., the zero function) if and only if $q = r$. An implication of this result is that the deviation of $\boldsymbol{g}^{q,r}$ from the zero function can be used as a measure of mismatch between $q$ and $r$. Different ways to characterize such deviation have led to different measures of goodness of fit.

The FSSD characterizes such deviation from $\boldsymbol{0}$ by evaluating $\boldsymbol{g}^{q,r}$ at $J_q$ test locations $W$. Formally, given a set of test locations $W = \{\boldsymbol{w}_j\}_{j=1}^{J_q}$, the squared FSSD is defined as $\mathrm{FSSD}_q^2(r) := \frac{1}{dJ_q} \sum_{j=1}^{J_q} \|\boldsymbol{g}^{q,r}(\boldsymbol{w}_j)\|_2^2 := F_q^2$ [Jitkrittum et al., 2017b]. Under appropriate conditions, it is known that almost surely $F_q^2 = 0$ if and only if $q = r$. Using the notations as in Jitkrittum et al. [2017b], one can write $F_q^2 = \mathbb{E}_{\boldsymbol{z} \sim r} \mathbb{E}_{\boldsymbol{z}' \sim r} \Delta_q(\boldsymbol{z}, \boldsymbol{z}')$ where $\Delta_q(\boldsymbol{z}, \boldsymbol{z}') := \boldsymbol{\tau}_q^\top(\boldsymbol{z}) \boldsymbol{\tau}_q(\boldsymbol{z}')$, $\boldsymbol{\tau}_q(\boldsymbol{z}) := \mathrm{vec}(\boldsymbol{\Xi}^q(\boldsymbol{z})) \in \mathbb{R}^{dJ_q}$, $\mathrm{vec}(\boldsymbol{M})$ concatenates columns of $\boldsymbol{M}$ into a column vector, and $\boldsymbol{\Xi}^q(\boldsymbol{z}) \in \mathbb{R}^{d \times J_q}$ is defined such that $[\boldsymbol{\Xi}^q(\boldsymbol{z})]_{i,j} := \xi_i^q(\boldsymbol{z}, \boldsymbol{w}_j)/\sqrt{dJ_q}$ for $i = 1, \dots, d$ and $j = 1, \dots, J_q$. Equivalently, $F_q^2 = \|\boldsymbol{\mu}_q\|_2^2$ where $\boldsymbol{\mu}_q := \mathbb{E}_{\boldsymbol{z} \sim r}[\boldsymbol{\tau}_q(\boldsymbol{z})]$. Similar to the UME statistic described previously, given a sample $Z_n = \{\boldsymbol{z}_i\}_{i=1}^{n} \sim r$, an unbiased estimator of $F_q^2$, denoted by $\widehat{F_q^2}$ can be straightforwardly written as a second-order U-statistic, which can be computed in $\mathcal{O}(J_q n)$ time. It was shown in Jitkrittum et al. [2017b] that the test locations $W$ can be chosen by maximizing the test power of the goodness-of-fit test proposing $H_0 : q = r$ against $H_1 : q \neq r$, using $\widehat{F_q^2}$ as the statistic. We note that, unlike UME, $\widehat{F_q^2}$ requires access to the density $q$. Another way to characterize the deviation of $\boldsymbol{g}^{q,r}$ from the zero function is to use the norm in the reproducing kernel Hilbert space (RKHS) that contains $\boldsymbol{g}^{q,r}$. This measure is known as the Kernel Stein Discrepancy having a runtime complexity of $\mathcal{O}(n^2)$ [Chwialkowski et al., 2016, Liu et al., 2016, Gorham and Mackey, 2015].

## 3   Proposal: Rel-UME and Rel-FSSD Tests

**Relative UME** (Rel-UME) Our first proposed relative goodness-of-fit test based on UME tests $H_0 : U^2(P, R) \leq U^2(Q, R)$ versus $H_1 : U^2(P, R) > U^2(Q, R)$. The test uses $\sqrt{n} \hat{S}_n^U = \sqrt{n} (\widehat{U_P^2} -$

$\widehat{U_Q^2}$) as the statistic, and rejects $H_0$ when it is larger than the threshold $T_\alpha$. The threshold is given by the $(1-\alpha)$-quantile of the asymptotic distribution of $\sqrt{n}\hat{S}_n^U$ when $H_0$ holds i.e., the null distribution, and the pre-chosen $\alpha$ is the significance level. It is well-known that this choice for the threshold asymptotically controls the false rejection rate to be bounded above by $\alpha$ yielding a level-$\alpha$ test [Casella and Berger, 2002, Definition 8.3.6]. In the full generality of Rel-UME, two sets of test locations can be used: $V = \{v_j\}_{j=1}^{J_p}$ for computing $\widehat{U_P^2}$, and $W = \{w_j\}_{j=1}^{J_q}$ for $\widehat{U_Q^2}$. The feature function for $\widehat{U_P^2}$ is denoted by $\psi_V(x) := \frac{1}{\sqrt{J_p}}\left(k_X(x, v_1), \dots, k_X(x, v_{J_p})\right)^\top \in \mathbb{R}^{J_p}$, for some kernel $k_X$ which can be different from $k_Y$ used in $\psi_W$. The asymptotic distribution of the statistic is stated in Theorem 1.

**Theorem 1** (Asymptotic distribution of $\hat{S}_n^U$). *Define $C_W^Q := \mathrm{cov}_{y \sim Q}[\psi_W(y), \psi_W(y)]$, $C_V^P := \mathrm{cov}_{x \sim P}[\psi_V(x), \psi_V(x)]$, and $C_{VW}^R := \mathrm{cov}_{z \sim R}[\psi_V(z), \psi_W(z)] \in \mathbb{R}^{J_p \times J_q}$. Let $S^U := U_P^2 - U_Q^2$, and $M := \begin{pmatrix} \psi_V^P - \psi_V^R & \mathbf{0} \\ \mathbf{0} & \psi_W^Q - \psi_W^R \end{pmatrix} \in \mathbb{R}^{(J_p+J_q) \times 2}$. Assume that 1) $P, Q$ and $R$ are all distinct, 2) $(k_X, V)$ are chosen such that $U_P^2 > 0$, and $(k_Y, W)$ are chosen such that $U_Q^2 > 0$, 3) $\begin{pmatrix} \zeta_P^2 & \zeta_{PQ} \\ \zeta_{PQ} & \zeta_Q^2 \end{pmatrix} := M^\top \begin{pmatrix} C_V^P + C_V^R & C_{VW}^R \\ (C_{VW}^R)^\top & C_W^Q + C_W^R \end{pmatrix} M$ is positive definite. Then,*
$$\sqrt{n}\left(\widehat{S}_n^U - S^U\right) \xrightarrow{d} \mathcal{N}\left(0, 4(\zeta_P^2 - 2\zeta_{PQ} + \zeta_Q^2)\right)$$

A proof of Theorem 1 can be found in Section C.1 (appendix). Let $\nu := 4(\zeta_P^2 - 2\zeta_{PQ} + \zeta_Q^2)$. Theorem 1 states that the asymptotic distribution of $\hat{S}_n^U$ is normal with the mean given by $S^U := U_P^2 - U_Q^2$. It follows that under $H_0$, $S^U \leq 0$ and the $(1-\alpha)$-quantile is $S^U + \sqrt{\nu}\Phi^{-1}(1-\alpha)$ where $\Phi^{-1}$ is the quantile function of the standard normal distribution. Since $S^U$ is unknown in practice, we therefore adjust it to be $\sqrt{\nu}\Phi^{-1}(1-\alpha)$, and use it as the test threshold $T_\alpha$. The adjusted threshold can be estimated easily by replacing $\nu$ with $\hat{\nu}_n$, a consistent estimate based on samples. It can be shown that the test with the adjusted threshold is still level-$\alpha$ (more conservative in rejecting $H_0$). We note that the same approach of adjusting the threshold is used in Rel-MMD [Bounliphone et al., 2015].

**Better Fit of $Q$ in Terms of $W$.** When specifying $V$ and $W$, the model comparison is done by comparing the goodness of fit of $P$ (to $R$) as measured in the regions specified by $V$ to the goodness of fit of $Q$ as measured in the regions specified by $W$. By specifying $V$ and setting $W = V$, testing with Rel-UME is equivalent to posing the question "*Does $Q$ fit to the data better than $P$ does, as measured in the regions of $V$?*" For instance, the observed sample from $R$ might contain smiling and non-smiling faces, and $P, Q$ are candidate generative models for face images. If we are interested in checking the relative fit in the regions of smiling faces, $V$ can be a set of smiling faces. In the followings, we will assume $V = W$ and $k := k_X = k_Y$ for interpretability. Investigating the general case without these constraints will be an interesting topic of future study. Importantly we emphasize that test results are always conditioned on the specified $V$. To be precise, let $U_V^2$ be the squared UME statistic defined by $V$. It is entirely realistic that the test rejects $H_0$ in favor of $H_1: U_{V_1}^2(P, R) > U_{V_1}^2(Q, R)$ (i.e., $Q$ fits better) for some $V_1$, and also rejects $H_0$ in favor of the opposite alternative $H_1: U_{V_2}^2(Q, R) > U_{V_2}^2(P, R)$ (i.e., $P$ fits better) for another setting of $V_2$. This is because the regions in which the model comparison takes place are different in the two cases. Although not discussed in Bounliphone et al. [2015], the same behaviour can be observed for Rel-MMD i.e., test results are conditioned on the choice of kernel.

In some cases, it is not known in advance what features are better represented by one model vs another, and it becomes necessary to learn these features from the model outputs. In this case, we propose setting $V$ to contain the locations which maximize the probability that the test can detect the better fit of $Q$, as measured at the locations. Following the same principle as in Gretton et al. [2012b], Sutherland et al. [2016], Jitkrittum et al. [2016, 2017a,b], this goal can be achieved by finding $(k, V)$ which maximize the test power, while ensuring that the test is level-$\alpha$. By Theorem 1, for large $n$ the test power $\mathbb{P}\left(\sqrt{n}\hat{S}_n^U > T_\alpha\right)$ is approximately $\Phi\left(\frac{\sqrt{n}S^U - T_\alpha}{\sqrt{\nu}}\right) = \Phi\left(\sqrt{n}\frac{S^U}{\sqrt{\nu}} - \sqrt{\frac{\hat{\nu}_n}{\nu}}\Phi^{-1}(1-\alpha)\right)$. Under $H_1$, $S^U > 0$. For large $n$, $\Phi^{-1}(1-\alpha)\sqrt{\hat{\nu}_n}/\sqrt{\nu}$ approaches a constant, and $\sqrt{n}S^U/\sqrt{\nu}$ dominates. It follows that, for large $n$, $(k^*, V^*) = \arg\max_{(k,V)} \mathbb{P}\left(\sqrt{n}\hat{S}_n^U > T_\alpha\right) \approx \arg\max_{(k,V)} S^U/\sqrt{\nu}$. We

can thus use $\hat{S}_n^U/(\gamma + \sqrt{\hat{\nu}_n})$ as an estimate of the *power criterion* objective $S^U/\sqrt{\nu}$ for the test power, where $\gamma > 0$ is a small regularization parameter added to promote numerical stability following Jitkrittum et al. [2017b, p. 5]. To control the false rejection rate, the maximization is carried out on held-out training data which are independent of the data used for testing. In the experiments (Section 4), we hold out 20% of the data for the optimization. A unique consequence of this procedure is that we obtain optimized $V^*$ which indicates where $Q$ fits significantly better than $P$. We note that this interpretation only holds if the test, using the optimized hyperparameters $(k^*, V^*)$, decides to reject $H_0$. The optimized locations may not be interpretable if the test fails to reject $H_0$.

**Relative FSSD** (Rel-FSSD) The proposed Rel-FSSD tests $H_0\colon F_p^2 \leq F_q^2$ versus $H_1\colon F_p^2 > F_q^2$. The test statistic is $\sqrt{n}\hat{S}_n^F := \sqrt{n}(\widehat{F_p^2} - \widehat{F_q^2})$. We note that the feature functions $\boldsymbol{\tau}_p$ (for $F_p^2$) and $\boldsymbol{\tau}_q$ (for $F_q^2$) depend on $(k_X, V)$ and $(k_Y, W)$ respectively, and play the same role as the feature functions $\psi_V$ and $\psi_W$ of the UME statistic. Due to space limitations, we only state the salient facts of Rel-FSSD. The rest of the derivations closely follow Rel-UME. These include the interpretation that the relative fit is measured at the specified locations given in $V$ and $W$, and the derivation of Rel-FSSD's power criterion (which can be derived using the asymptotic distribution of $\hat{S}_n^F$ given in Theorem 2, following the same line of reasoning as in the case of Rel-UME). A major difference is that Rel-FSSD requires explicit (gradients of the log) density functions of the two models, allowing it to gain structural information of the models that may not be as easily observed in finite samples. We next state the asymptotic distribution of the statistic (Theorem 2), which is needed for obtaining the threshold and for deriving the power criterion. The proof closely follows the proof of Theorem 1, and is omitted.

**Theorem 2** (Asymptotic distribution of $\hat{S}_n^F$). *Define* $S^F := F_p^2 - F_q^2$. *Let* $\boldsymbol{\Sigma}^{ss'} := \mathrm{cov}_{\boldsymbol{z} \sim r}[\boldsymbol{\tau}_s(\boldsymbol{z}), \boldsymbol{\tau}_{s'}(\boldsymbol{z})]$ *for* $s, s' \in \{p, q\}$ *so that* $\boldsymbol{\Sigma}^{pq} \in \mathbb{R}^{dJ_p \times dJ_q}$, $\boldsymbol{\Sigma}^{qp} := (\boldsymbol{\Sigma}^{pq})^\top$, $\boldsymbol{\Sigma}^{pp} = \boldsymbol{\Sigma}^p \in \mathbb{R}^{dJ_p \times dJ_p}$, *and* $\boldsymbol{\Sigma}^{qq} = \boldsymbol{\Sigma}^q \in \mathbb{R}^{dJ_q \times dJ_q}$. *Assume that 1) $p, q$, and $r$ are all distinct, 2) $(k_X, V)$ are chosen such that $F_p^2 > 0$, and $(k_Y, W)$ are chosen such that $F_q^2 > 0$, 3)*
$$\begin{pmatrix} \sigma_p^2 & \sigma_{pq} \\ \sigma_{pq} & \sigma_q^2 \end{pmatrix} := \begin{pmatrix} \boldsymbol{\mu}_p^\top \boldsymbol{\Sigma}^p \boldsymbol{\mu}_p & \boldsymbol{\mu}_p^\top \boldsymbol{\Sigma}^{pq} \boldsymbol{\mu}_q \\ \boldsymbol{\mu}_p^\top \boldsymbol{\Sigma}^{pq} \boldsymbol{\mu}_q & \boldsymbol{\mu}_q^\top \boldsymbol{\Sigma}^q \boldsymbol{\mu}_q \end{pmatrix} \text{ is positive definite. Then, } \sqrt{n}\left(\hat{S}_n^F - S^F\right) \xrightarrow{d}$$
$\mathcal{N}\left(0, 4(\sigma_p^2 - 2\sigma_{pq} + \sigma_q^2)\right)$.

# 4 Experiments

In this section, we demonstrate the two proposed tests on both toy and real problems. We start with an illustration of the behaviors of Rel-UME and Rel-FSSD's power criteria using simple one-dimensional problems. In the second experiment, we examine the test powers of the two proposed tests using three toy problems. In the third experiment, we compare two hypothetical generative models on the CIFAR-10 dataset [Krizhevsky and Hinton, 2009] and demonstrate that the learned test locations (images) can clearly indicate the types of images that are better modeled by one of the two candidate models. In the last two experiments, we consider the problem of determining the relative goodness of fit of two given Generative Adversarial Networks (GANs) [Goodfellow et al., 2014]. Code to reproduce all the results is available at `https://github.com/wittawatj/kernel-mod`.

**1. Illustration of** Rel-UME **and** Rel-FSSD **Power Criteria** We consider $k = k_X = k_Y$ to be a Gaussian kernel, and set $V = W = \{\boldsymbol{v}\}$ (one test location). The power criterion of Rel-UME as a function of $\boldsymbol{v}$ can be written as $\frac{1}{2} \frac{\mathrm{wit}_{P,R}^2(\boldsymbol{v}) - \mathrm{wit}_{Q,R}^2(\boldsymbol{v})}{(\zeta_P^2(\boldsymbol{v}) - 2\zeta_{PQ}(\boldsymbol{v}) + \zeta_Q^2(\boldsymbol{v}))^{1/2}}$ where $\mathrm{wit}(\cdot)$ is the MMD witness function (see Section 2), and we explicitly indicate the dependency on $\boldsymbol{v}$. To illustrate, we consider two Gaussian models $p, q$ with different means but the same variance, and set $r$ to be a mixture of $p$ and $q$. Figure 1a shows that when each component in $r$ has the same mixing proportion, the power criterion of Rel-UME is a zero function indicating that $p$ and $q$ have the same goodness of fit to $r$ everywhere. To understand this, notice that at the left mode of $r$, $p$ has excessive probability mass (compared to $r$), while $q$ has almost no mass at all. Both models are thus wrong at the left mode of $r$. However, since the extra probability mass of $p$ is equal to the missing mass of $q$, Rel-UME considers $p$ and $q$ as having the same goodness of fit. In Figure 1b, the left mode of $r$ now has a mixing proportion of only 30%, and $r$ more closely matches $q$. The power criterion is thus positive at the left mode indicating that $q$ has a better fit.

The power criterion of Rel-FSSD indicates that $q$ fits better at the right mode of $r$ in the case of equal mixing proportion (see Figure 1c). In one dimension, the Stein witness function $\boldsymbol{g}^{q,r}$ (defined

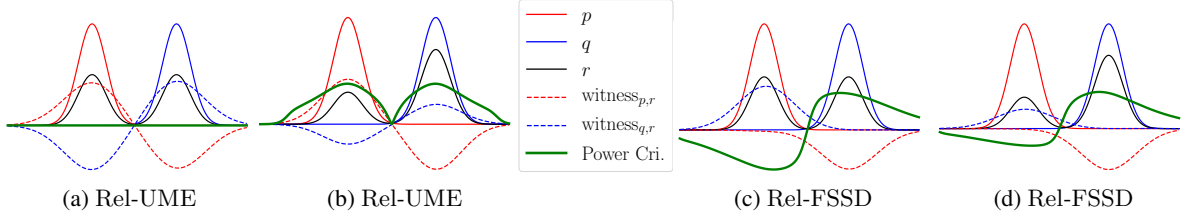

| | |
|---|---|
| —— | $p$ |
| —— | $q$ |
| —— | $r$ |
| - - - | $\text{witness}_{p,r}$ |
| - - - | $\text{witness}_{q,r}$ |
| —— | Power Cri. |

(a) Rel-UME      (b) Rel-UME      (c) Rel-FSSD      (d) Rel-FSSD

Figure 1: One-dimensional plots (in green) of Rel-UME's power criteria (in (a), (b)), and Rel-FSSD's power criteria (in (c), (d)). The dashed lines in (a), (b) indicate MMD's witness functions used in Rel-UME, and the dashed lines in (c), (d) indicate FSSD's Stein witness functions.

in Section 2) can be written as $g^{q,r}(w) = \mathbb{E}_{z \sim r} \left[ k_Y(z, w) \nabla_z (\log q(z) - \log r(z)) \right]$, which is the expectation under $r$ of the difference in the derivative log of $q$ and $r$, weighted by the kernel $k_Y$. The Stein witness thus only captures the matching of the shapes of the two densities (as given by the derivative log). Unlike the MMD witness, the Stein witness is insensitive to the mismatch of probability masses i.e., it is independent of the normalizer of $q$. In Figure 1c, since the shape of $q$ and the shape of the right mode of $r$ match, the Stein witness $g^{q,r}$ (dashed blue curve) vanishes at the right mode of $r$, indicating a good fit of $q$ in the region. The mismatch between the shape of $q$ and the shape of $r$ at the left mode of $r$ is what creates the peak of $g^{q,r}$. The same reasoning holds for the Stein witness $g^{p,r}$. The power criterion of Rel-FSSD, which is given by $\frac{1}{2} \frac{g^{p,r}(w)^2 - g^{q,r}(w)^2}{(\sigma_p^2(w) - 2\sigma_{pq}(w) + \sigma_q^2(w))^{1/2}}$, is thus positive at the right mode of $r$ (shapes of $q$ and $r$ matched there), and negative at the left mode of $r$ (shapes of $p$ and $r$ matched there). To summarize, Rel-UME measures the relative fit by checking the probability mass, while Rel-FSSD does so by matching the shapes of the densities.

**2. Test Powers on Toy Problems** The goal of this experiment is to investigate the rejection rates of several variations of the two proposed tests. To this end, we study three toy problems, each having its own characteristics. All the three distributions in each problem have density functions to allow comparison with Rel-FSSD.

1. *Mean shift*: All the three distributions are isotropic multivariate normal distributions: $p = \mathcal{N}([0.5, 0, \ldots, 0], \boldsymbol{I})$, $q = \mathcal{N}([1, 0, \ldots 0], \boldsymbol{I})$, and $r = \mathcal{N}(\boldsymbol{0}, \boldsymbol{I})$, defined on $\mathbb{R}^{50}$. The two candidates models $p$ and $q$ differ in the mean of the first dimension. In this problem, the null hypothesis $H_0$ is true since $p$ is closer to $r$.

2. *Blobs*: Each distribution is given by a mixture of four Gaussian distributions organized in a grid in $\mathbb{R}^2$. Samples from $p, q$ and $r$ are shown in Figure 4. In this problem, $q$ is closer to $r$ than $p$ is i.e., $H_1$ is true. One characteristic of this problem is that the difference between $p$ and $q$ takes place in a small scale relative to the global structure of the data. This problem was studied in Gretton et al. [2012b], Chwialkowski et al. [2015].

3. *RBM*: Each of the three distributions is given by a Gaussian Bernoulli Restricted Boltzmann Machine (RBM) model with density function $p'_{\boldsymbol{B}, \boldsymbol{b}, \boldsymbol{c}}(\boldsymbol{x}) = \sum_{\boldsymbol{h}} p'_{\boldsymbol{B}, \boldsymbol{b}, \boldsymbol{c}}(\boldsymbol{x}, \boldsymbol{h})$, where $p'_{\boldsymbol{B}, \boldsymbol{b}, \boldsymbol{c}}(\boldsymbol{x}, \boldsymbol{h}) := \frac{1}{Z} \exp\left( \boldsymbol{x}^\top \boldsymbol{B} \boldsymbol{h} + \boldsymbol{b}^\top \boldsymbol{x} + \boldsymbol{c}^\top \boldsymbol{h} - \frac{1}{2} \|\boldsymbol{x}\|^2 \right)$, $\boldsymbol{h} \in \{-1, 1\}^{d_h}$ is a latent vector, $Z$ is the normalizer, and $\boldsymbol{B}, \boldsymbol{b}, \boldsymbol{c}$ are model parameters. Let $r(\boldsymbol{x}) := p'_{\boldsymbol{B}, \boldsymbol{b}, \boldsymbol{c}}(\boldsymbol{x}), p(\boldsymbol{x}) := p'_{\boldsymbol{B}^p, \boldsymbol{b}, \boldsymbol{c}}(\boldsymbol{x})$, and $q(\boldsymbol{x}) := p'_{\boldsymbol{B}^q, \boldsymbol{b}, \boldsymbol{c}}(\boldsymbol{x})$. Following a similar setting as in Liu et al. [2016], Jitkrittum et al. [2017b], we set the parameters of the data generating density $r$ by uniformly randomly setting entries of $\boldsymbol{B}$ to be from $\{-1, 1\}$, and drawing entries of $\boldsymbol{b}$ and $\boldsymbol{c}$ from the standard normal distribution. Let $\boldsymbol{\delta}$ be a matrix of the same size as $\boldsymbol{B}$ such that $\delta_{1,1} = 1$ and all other entries are 0. We set $\boldsymbol{B}^q = \boldsymbol{B} + 0.3\boldsymbol{\delta}$ and $\boldsymbol{B}^p = \boldsymbol{B} + \epsilon\boldsymbol{\delta}$, where the perturbation constant $\epsilon$ is varied. We fix the sample size $n$ to 2000. Perturbing only one entry of $\boldsymbol{B}$ creates a problem in which the difference of distributions can be difficult to detect. This serves as a challenging benchmark to measure the sensitivity of statistical tests [Jitkrittum et al., 2017b]. We set $d = 20$ and $d_h = 5$.

We compare three kernel-based tests: Rel-UME, Rel-FSSD, and Rel-MMD (the relative MMD test of Bounliphone et al. [2015]), all using a Gaussian kernel. For Rel-UME and Rel-FSSD we set $k_X = k_Y = k$, where the the Gaussian width of $k$, and the test locations are chosen by maximizing their respective power criteria described in Section 3 on 20% of the data. The optimization procedure is described in Section A (appendix). Following Bounliphone et al. [2015], the Gaussian width of Rel-MMD is chosen by the median heuristic as implemented in the code by the authors. In the RBM

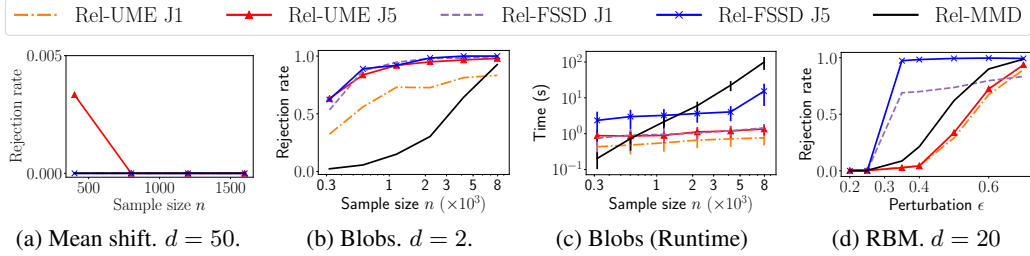

| --·- Rel-UME J1 | —▲— Rel-UME J5 | ---- Rel-FSSD J1 | —✕— Rel-FSSD J5 | —— Rel-MMD |

(a) Mean shift. $d = 50$.  (b) Blobs. $d = 2$.  (c) Blobs (Runtime)  (d) RBM. $d = 20$

Figure 2: (a), (b), (d) Rejection rates (estimated from 300 trials) of the five tests with $\alpha = 0.05$. In the RBM problem, $n = 2000$. (c) Runtime in seconds for one trial in the Blobs problem.

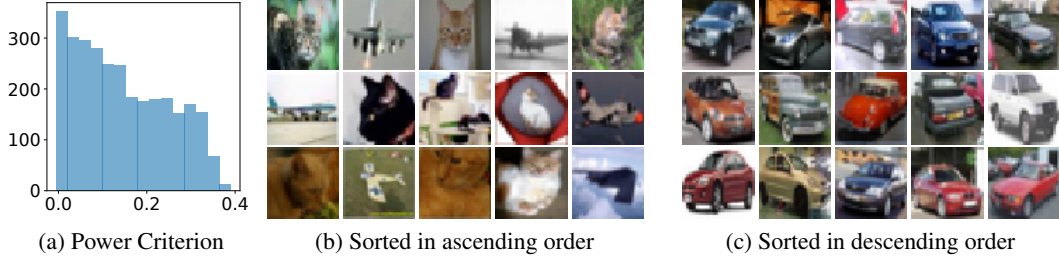

(a) Power Criterion  (b) Sorted in ascending order  (c) Sorted in descending order

Figure 3: $P = \{\text{airplane, cat}\}$, $Q = \{\text{automobile, cat}\}$, and $R = \{\text{automobile, cat}\}$. (a) Histogram of Rel-UME power criterion values. (b), (c) Images as sorted by the criterion values in ascending and descending orders, respectively.

problem, all problem parameters $B$, $b$, and $c$ are drawn only once and fixed. Only the samples vary across trials.

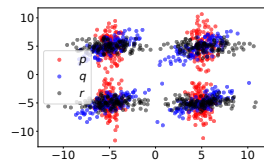

Figure 4: Blobs problem samples: $p, q, r$.

Figure 2 shows the test powers of all the tests. When $H_0$ holds, all tests have false rejection rates (type-I errors) bounded above by $\alpha = 0.05$ (Figure 2a). In the Blobs problem (Figure 2b), it can be seen that Rel-UME achieves larger power at all sample sizes, compared to Rel-MMD. Since the relative goodness of fit of $p$ and $q$ must be compared locally, the optimized test locations of Rel-UME are suitable for detecting such local differences. The poor performance of Rel-MMD is caused by unsuitable choices of the kernel bandwidth. The bandwidth chosen by the median heuristic is only appropriate for capturing the global length scale of the problem. It is thus too large to capture small-scale differences. No existing work has proposed a kernel selection procedure for Rel-MMD. Regarding the number $J$ of test locations, we observe that changing $J$ from 1 to 5 drastically increases the test power of Rel-UME, since more regions characterizing the differences can be pinpointed. Rel-MMD exhibits a quadratic-time profile (Figure 2c) as a function of $n$.

Figure 2d shows the rejection rates against the perturbation strength $\epsilon$ in $p$ in the RBM problem. When $\epsilon \le 0.3$, $p$ is closer to $r$ than $q$ is (i.e., $H_0$ holds). We observe that all the tests have well-controlled false rejection rates in this case. At $\epsilon = 0.35$, while $q$ is closer (i.e., $H_1$ holds), the relative amount by which $q$ is closer to $r$ is so small that a significant difference cannot be detected when $p$ and $q$ are represented by samples of size $n = 2000$, hence the low powers of Rel-UME and Rel-MMD. Structural information provided by the density functions allows Rel-FSSD (both $J = 1$ and $J = 5$) to detect the difference even at $\epsilon = 0.35$, as can be seen from the high test powers. The fact that Rel-MMD has higher power than Rel-UME, and the fact that changing $J$ from 1 to 5 increases the power only slightly suggest that the differences may be spatially diffuse (rather than local).

**3. Informative Power Objective** In this part, we demonstrate that test locations having positive (negative) values of the power criterion correctly indicate the regions in which $Q$ has a better (worse) fit. We consider image samples from three categories of the CIFAR-10 dataset [Krizhevsky and Hinton, 2009]: airplane, automobile, and cat. We partition the images, and assume that the sample from $P$ consists of 2000 airplane, 1500 cat images, the sample from $Q$ consists of 2000 automobile, 1500 cat images, and the reference sample from $R$ consists of 2000 automobile, 1500 cat images. All samples are independent. We consider a held-out random sample consisting of 1000 images from each

Table 1: Rejection rates of the proposed Rel-UME, Rel-MMD, KID and FID, in the GAN model comparison problem. "FID diff." refers to the average of $\mathrm{FID}(P, R) - \mathrm{FID}(Q, R)$ estimated in each trial. Significance level $\alpha = 0.01$ (for Rel-UME, Rel-MMD, and KID).

| | $P$ | $Q$ | $R$ | Rel-UME | | | Rel-MMD | KID | FID | FID diff. |
|---|---|---|---|---|---|---|---|---|---|---|
| | | | | J10 | J20 | J40 | | | | |
| 1. | S | S | RS | 0.0 | 0.0 | 0.0 | 0.0 | 0.0 | 0.53 | $-0.045 \pm 0.52$ |
| 2. | RS | RS | RS | 0.0 | 0.0 | 0.0 | 0.03 | 0.02 | 0.7 | $0.04 \pm 0.19$ |
| 3. | S | N | RS | 0.0 | 0.0 | 0.0 | 0.0 | 0.0 | 0.0 | $-15.22 \pm 0.83$ |
| 4. | S | N | RN | 0.57 | 0.97 | 1.0 | 1.0 | 1.0 | 1.0 | $5.25 \pm 0.75$ |
| 5. | S | N | RM | 0.0 | 0.0 | 0.0 | 0.0 | 0.0 | 0.0 | $-4.55 \pm 0.82$ |

category, serving as a pool of test location candidates. We set the kernel to be the Gaussian kernel on 2048 features extracted by the Inception-v3 network at the pool3 layer [Szegedy et al., 2016]. We evaluate the power criterion of Rel-UME at each of the test locations in the pool individually. The histogram of the criterion values is shown in Figure 3a. We observe that all the power criterion values are non-negative, confirming that $Q$ is better than $P$ everywhere. Figure 3b shows the top 15 test locations as sorted in ascending order by the criterion, consisting of automobile images. These indicate the regions in the data domain where $Q$ fits better. Notice that cat images do not have high positive criterion values because they can be modeled equally well by $P$ and $Q$, and thus have scores close to zero as shown in Figure 3b.

**4. Testing GAN Models** In this experiment, we apply the proposed Rel-UME test to comparing two generative adversarial networks (GANs) [Goodfellow et al., 2014]. We consider the CelebA dataset [Liu et al., 2015][3] in which each data point is an image of a celebrity with 40 binary attributes annotated e.g., pointy nose, smiling, mustache, etc. We create a partition of the images on the *smiling* attribute, thereby creating two disjoint subsets of *smiling* and *non-smiling* images. A set of 30000 images from each subset is held out for subsequent relative goodness-of-fit testing, and the rest are used for training two GAN models: a model for smiling images, and a model for non-smiling images. Generated samples and details of the trained models can be found in Section B (appendix). The two models are trained once and fixed throughout.

In addition to Rel-MMD, we compare the proposed Rel-UME to Kernel Inception Distance (KID) [Bińkowski et al., 2018], and Fréchet Inception Distance (FID) [Heusel et al., 2017], which are distances between two samples (originally proposed for comparing a sample of generated images, and a reference sample). All images are represented by 2048 features extracted from the Inception-v3 network [Szegedy et al., 2016] at the pool3 layer following Bińkowski et al. [2018]. When adapted for three samples, KID is in fact a variant of Rel-MMD in which a third-order polynomial kernel is used instead of a Gaussian kernel (on top of the pool3 features). Following Bińkowski et al. [2018], we construct a bootstrap estimator for FID (10 subsamples with 1000 points in each). For the proposed Rel-UME, the $J \in \{10, 20, 40\}$ test locations are randomly set to contain $J/2$ smiling images, and $J/2$ non-smiling images drawn from a held-out set of real images. We create problem variations by setting $P, Q, R \in \{S, N, RS, RN, RM\}$ where S denotes generated smiling images (from the trained model), N denotes generated non-smiling images, M denotes an equal mixture of smiling and non-smiling images, and the prefix R indicates that real images are used (as opposed to generated ones). The sample size is $n = 2000$, and each problem variation is repeated for 10 trials for FID (due to its high complexity) and 100 trials for other methods. The rejection rates from all the methods are shown in Table 1. Here, the test result for FID in each trial is considered "reject $H_0$" if $\mathrm{FID}(P, R) > \mathrm{FID}(Q, R)$. Heusel et al. [2017] did not propose FID as a statistical test. That said, there is a generic way of constructing a relative goodness-of-fit test based on repeated permutation of samples of $P$ and $Q$ to simulate from the null distribution. However, FID requires computing the square root of the feature covariance matrix (2048 x 2048), and is computationally too expensive for permutation testing.

Overall, we observe that the proposed test does at least equally well as existing approaches, in identifying the better model in each case. In problems 1 and 2, $P$ and $Q$ have the same goodness of fit, by design. In these cases, all the tests correctly yield low rejection rates, staying roughly at the design level ($\alpha = 0.01$). Without a properly chosen threshold, the (false) rejection rates of FID fluctuate

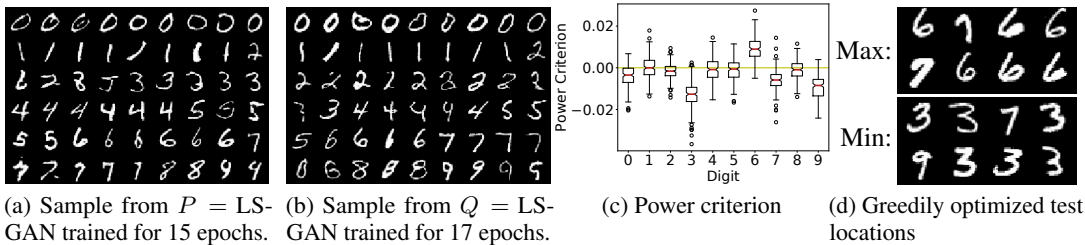

(a) Sample from $P = $ LS-GAN trained for 15 epochs.    (b) Sample from $Q = $ LS-GAN trained for 17 epochs.    (c) Power criterion    (d) Greedily optimized test locations

Figure 5: Examining the training of an LSGAN model with Rel-UME. (a), (b) Samples from the two models $P, Q$ trained on MNIST. (c) Distributions of power criterion values computed over 200 trials. Each distribution is formed by randomly selecting $J = 40$ test locations from real images of a digit type. (d) Test locations showing where $Q$ is better (maximization of the power criterion), and test locations showing where $P$ is better (minimization).

around the expected value of 0.5. This means that simply comparing FIDs (or other distances) to the reference sample without a calibrated threshold can lead to a wrong conclusion on the relative goodness of fit. The FID is further complicated by the fact that its estimator suffers from bias in ways that are hard to model and correct for (see Bińkowski et al. [2018, Section D.1]). Problem 4 is a case where the model $Q$ is better. We notice that increasing the number of test locations of Rel-UME helps detect the better fit of $Q$. In problem 5, the reference sample is bimodal, and each model can capture only one of the two modes (analogous to the synthetic problem in Figure 1a). All the tests correctly indicate that no model is better than another.

**5. Examining GAN Training** In the final experiment, we show that the power criterion of Rel-UME can be used to examine the relative change of the distribution of a GAN model after training further for a few epochs. To illustrate, we consider training an LSGAN model [Mao et al., 2017] on MNIST, a dataset in which each data point is an image of a handwritten digit. We set $P$ and $Q$ to be LSGAN models after 15 epochs and 17 epochs of training, respectively. Details regarding the network architecture, training, and the kernel (chosen to be a Gaussian kernel on features extracted from a convolutional network) can be found in Section D. Samples from $P$ and $Q$ are shown in Figures 5a and 5b (see Figure 8 in the appendix for more samples).

We set the test locations $V$ to be the set $V_i$ containing $J = 40$ randomly selected real images of digit $i$, for $i \in \{0, \dots, 9\}$. We then draw $n = 2000$ points from $P, Q$ and the real data $(R)$, and use $V = V_i$ to compute the power criterion for $i \in \{0, \dots, 9\}$. The procedure is repeated for 200 trials where $V$ and the samples are redrawn each time. The results are shown in Figure 5c. We observe that when $V = V_3$ (i.e., box plot at the digit 3) or $V_9$, the power criterion values are mostly negative, indicating that $P$ is better than $Q$, as measured in the regions indicated by real images of the digits 3 or 9. By contrast, when $V = V_6$, the large mass of the box plot in the positive orthant shows that $Q$ is better in the regions of the digit 6. For other digits, the criterion values spread around zero, showing that there is no difference between $P$ and $Q$, on average. We further confirm that the class proportions of the generated digits from both models are roughly correct (i.e., uniform distribution), meaning that the difference between $P$ and $Q$ in these cases is not due to the mismatch in class proportions (see Section D). These observations imply that after the 15th epoch, training this particular LSGAN model two epochs further improves generation of the digit 6, and degrades generation of digits 3 and 9. A non-monotonic improvement during training is not uncommon since at the 15th epoch the training has not converged. More experimental results from comparing different GAN variants on MNIST can be found in Section E in the appendix.

We note that the set $V$ does not need to contain test locations of the same digit. In fact, the notion of class labels may not even exist in general. It is up to the user to define $V$ to contain examples which capture the relevant concept of interest. For instance, to compare the ability of models to generate straight strokes, one might include digits 1 and 7 in the set $V$. An alternative to manual specification of $V$ is to optimize the power criterion to find the locations that best distinguish the two models (as done in experiment 2). To illustrate, we consider greedily optimizing the power criterion by iteratively selecting a test location (from real images) which best improves the objective. Maximizing the objective yields locations that indicate the better fit of $Q$, whereas minimization gives locations which show the better fit of $P$ (recall from Figure 1). The optimized locations are shown in Figure 5d. The results largely agree with our previous observations, and do not require manually specifying $V$. This optimization procedure is applicable to any models which can be sampled.

**Acknowledgments**

HK and AG thank the Gatsby Charitable Foundation for the financial support.

## Footnotes

[2]In this work, since the distance is always measured relative to the data generating distribution $R$, we write $U_Q$ instead of $U(Q, R)$ to avoid cluttering the notation.

[3]CelebA dataset: http://mmlab.ie.cuhk.edu.hk/projects/CelebA.html.

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
