[Supplementary Material]

# Informative Features for Model Comparison
## Supplementary

## A  Optimization of Test Locations in Rel-UME and Rel-FSSD

This section describes the optimization procedure we use to select the test locations $V$ and the bandwidth of the Gaussian kernel in the experiment "Test Powers on Toy Problems." Since the two sets $V, W$ of test locations are constrained to be the same i.e., $V = W$ consisting of $J = J_p = J_q$ locations, in total, we have $Jd+1$ parameters. We follow a similar implementation of the optimization procedure for finding the test locations in FSSD.[4] All the parameters are optimized jointly by gradient ascent. We initialize the test locations by randomly picking $J$ points from the training set. The Gaussian width is initialized (for gradient ascent) to the square of the mean of $\mathrm{med}_{X^{tr} \cup Z^{tr}}$ and $\mathrm{med}_{Y^{tr} \cup Z^{tr}}$, where $\mathrm{med}_A := \mathrm{median}\left( \{\|\boldsymbol{x} - \boldsymbol{x}'\|_2\}_{\boldsymbol{x}, \boldsymbol{x}' \in A} \right)$. This is a similar heuristic used in Bounliphone et al. [2015] to set the bandwidth of the Gaussian kernel for Rel-MMD.

## B  Trained Models for Generating Smiling and Non-Smiling Images

(a) Samples from the smiling model            (b) Samples from the non-smiling model

Figure 6: Samples from the two trained models (smiling, and non-smiling) used in "Testing GAN Models" experiment in Section 4.

This section describes the details of the two GAN models (smiling, and non-smiling models) we use in the "Testing GAN Models" experiment in Section 4. We use the CelebA dataset [Liu et al., 2015] in which each data point is an image of a celebrity with 40 binary attributes annotated e.g., pointy nose, smiling, mustache, etc. We create a partition of the images on the *smiling* attribute, thereby creating two disjoint subsets of *smiling* and *non-smiling* images. To reduce confounding factors that are not related to smiling (e.g., sunglasses, background), each image is cropped to be 64x64 pixels, so that only the face remains. Cropping and image alignment with eyes and lips are done with the software described in Amos et al. [2016]. We use DCGAN architecture [Radford et al., 2015] (for both generator and discriminator) for both smiling and non-smiling models, coded in Pytorch. Subsampling was performed so that the training sizes for the two models are equal. Each model is trained on 84,822 images (i.e., 84822 smiling faces, and 84822 non-smiling faces) for 50 epochs. The training time was roughly three hours using an Nvidia Titan X graphics card with Pascal architecture. We use Adam optimizer [Kingma and Ba, 2014] with $\beta_1 = 0.5$ and $\beta_2 = 0.999$. The learning rate is set to $10^{-3}$ (for both discriminator and generator in the two models). Some samples generated from the two trained models are shown in Figure 6.

## C  Proofs

This section contains proofs for the results given in the main text.

### C.1  Proof of Theorem 1

Let all the notations be defined as in Section 3. Recall Theorem 1:

**Theorem 1** (Asymptotic distribution of $\hat{S}_n^U$). *Define* $C_W^Q := \mathrm{cov}_{\boldsymbol{y} \sim Q}[\psi_W(\boldsymbol{y}), \psi_W(\boldsymbol{y})]$, $C_V^P :=$ $\mathrm{cov}_{\boldsymbol{x} \sim P}[\psi_V(\boldsymbol{x}), \psi_V(\boldsymbol{x})]$, *and* $C_{VW}^R := \mathrm{cov}_{\boldsymbol{z} \sim R}[\psi_V(\boldsymbol{z}), \psi_W(\boldsymbol{z})] \in \mathbb{R}^{J_p \times J_q}$. *Let* $S^U := U_P^2 - U_Q^2$, *and* $\boldsymbol{M} := \begin{pmatrix} \psi_V^P - \psi_V^R & \boldsymbol{0} \\ \boldsymbol{0} & \psi_W^Q - \psi_W^R \end{pmatrix} \in \mathbb{R}^{(J_p + J_q) \times 2}$. *Assume that 1) $P, Q$ and $R$ are all distinct, 2) $(k_X, V)$ are chosen such that $U_P^2 > 0$, and $(k_Y, W)$ are chosen such that $U_Q^2 > 0$, 3) $\begin{pmatrix} \zeta_P^2 & \zeta_{PQ} \\ \zeta_{PQ} & \zeta_Q^2 \end{pmatrix} := \boldsymbol{M}^\top \begin{pmatrix} C_V^P + C_V^R & C_{VW}^R \\ (C_{VW}^R)^\top & C_W^Q + C_W^R \end{pmatrix} \boldsymbol{M}$ is positive definite. Then,*
$$\sqrt{n}\left(\widehat{S}_n^U - S^U\right) \xrightarrow{d} \mathcal{N}\left(0, 4(\zeta_P^2 - 2\zeta_{PQ} + \zeta_Q^2)\right).$$

*Proof.* Consider a random vector $\boldsymbol{t} := (\boldsymbol{x}, \boldsymbol{y}, \boldsymbol{z}) \in \mathcal{X}^3$, where $\boldsymbol{x}, \boldsymbol{y}$, and $\boldsymbol{z}$ are independently drawn from $P, Q$, and $R$, respectively. Let $T$ be the distribution of $\boldsymbol{t}$, and $\{\boldsymbol{t}_i\}_{i=1}^n = \{(\boldsymbol{x}_i, \boldsymbol{y}_i, \boldsymbol{z}_i)\}_{i=1}^n \overset{i.i.d.}{\sim} T$. Define two functions
$$\delta_V^P(\boldsymbol{t}, \boldsymbol{t}') := (\psi_V(\boldsymbol{x}) - \psi_V(\boldsymbol{z}))^\top (\psi_V(\boldsymbol{x}') - \psi_V(\boldsymbol{z}')),$$
$$\delta_W^Q(\boldsymbol{t}, \boldsymbol{t}') := (\psi_W(\boldsymbol{y}) - \psi_W(\boldsymbol{z}))^\top (\psi_W(\boldsymbol{y}') - \psi_W(\boldsymbol{z}')),$$
where $\boldsymbol{t}' := (\boldsymbol{x}', \boldsymbol{y}', \boldsymbol{z}')$. It can be seen that $\delta_V^P(\boldsymbol{t}, \boldsymbol{t}') = \delta_V^P(\boldsymbol{t}', \boldsymbol{t})$ and $\delta_W^Q(\boldsymbol{t}, \boldsymbol{t}') = \delta_W^Q(\boldsymbol{t}', \boldsymbol{t})$ for all $\boldsymbol{t}, \boldsymbol{t}' \in \mathcal{X}^3$, and that both functions are valid U-statistic kernels. It is not difficult to see that $\widehat{U_P^2}$ and $\widehat{U_Q^2}$ (estimator given in Section 2) can be written in the form of second-order U-statistics [Serfling, 2009, Chapter 5] as
$$\widehat{U_P^2} = \binom{n}{2}^{-1} \sum_{i=1}^n \sum_{j<i} \delta_V^P(\boldsymbol{t}, \boldsymbol{t}'),$$
$$\widehat{U_Q^2} = \binom{n}{2}^{-1} \sum_{i=1}^n \sum_{j<i} \delta_W^Q(\boldsymbol{t}, \boldsymbol{t}').$$

Since $\psi_V^P \neq \psi_V^R$ (because $U_P^2 > 0$), $\widehat{U_P^2}$ is a non-degenerate U-statistic. Since $\psi_W^Q \neq \psi_W^R$, $\widehat{U_Q^2}$ is also non-degenerate [Serfling, 2009, Section 5.5.1]. By Hoeffding [1948, Theorem 7.1], asymptotically their joint distribution is given by a normal distribution:
$$\sqrt{n}\left(\begin{pmatrix} \widehat{U_P^2} \\ \widehat{U_Q^2} \end{pmatrix} - \begin{pmatrix} U_P^2 \\ U_Q^2 \end{pmatrix}\right) \xrightarrow{d} \mathcal{N}\left(\boldsymbol{0}, 4\begin{pmatrix} \zeta_P^2 & \zeta_{PQ} \\ \zeta_{PQ} & \zeta_Q^2 \end{pmatrix}\right), \tag{1}$$

where
$$\zeta_P^2 = \mathbb{V}_{\boldsymbol{t} \sim T}\left[\mathbb{E}_{\boldsymbol{t}' \sim T}[\delta_V^P(\boldsymbol{t}, \boldsymbol{t}')]\right] \overset{(a)}{=} (\psi_V^P - \psi_V^R)^\top (C_V^P + C_V^R)(\psi_V^P - \psi_V^R),$$
$$\zeta_Q^2 = \mathbb{V}_{\boldsymbol{t} \sim T}\left[\mathbb{E}_{\boldsymbol{t}' \sim T}[\delta_W^Q(\boldsymbol{t}, \boldsymbol{t}')]\right] \overset{(b)}{=} (\psi_W^Q - \psi_W^R)^\top (C_W^Q + C_W^R)(\psi_W^Q - \psi_W^R),$$
$$\zeta_{PQ} = \mathrm{cov}_{\boldsymbol{t} \sim T}\left(\mathbb{E}_{\boldsymbol{t}' \sim T}[\delta_V^P(\boldsymbol{t}, \boldsymbol{t}')], \mathbb{E}_{\boldsymbol{t}' \sim T}[\delta_W^Q(\boldsymbol{t}, \boldsymbol{t}')]\right) \overset{(c)}{=} (\psi_V^P - \psi_V^R)^\top C_{VW}^R (\psi_W^Q - \psi_W^R),$$

and $C_{VW}^R := \mathrm{cov}_{\boldsymbol{z} \sim R}[\psi_V(\boldsymbol{z}), \psi_W(\boldsymbol{z})] \in \mathbb{R}^{J_p \times J_q}$. At $(a), (b), (c)$, we rely on the independence among $\boldsymbol{x}, \boldsymbol{y}$, and $\boldsymbol{z}$. A direct calculation gives the expressions of $\zeta_P^2, \zeta_Q^2$, and $\zeta_{PQ}$. By the continuous mapping theorem, and (1), $\sqrt{n}\begin{pmatrix} 1 \\ -1 \end{pmatrix}^\top \left(\begin{pmatrix} \widehat{U_P^2} \\ \widehat{U_Q^2} \end{pmatrix} - \begin{pmatrix} U_P^2 \\ U_Q^2 \end{pmatrix}\right) = \sqrt{n}\left(\widehat{S}_n^U - S^U\right) \xrightarrow{d}$
$$\mathcal{N}\left(\boldsymbol{0}, 4\begin{pmatrix} 1 \\ -1 \end{pmatrix}^\top \begin{pmatrix} \zeta_P^2 & \zeta_{PQ} \\ \zeta_{PQ} & \zeta_Q^2 \end{pmatrix}\begin{pmatrix} 1 \\ -1 \end{pmatrix}\right) \text{ giving the result.} \qquad \square$$

Table 2: Discriminator and generator of LSGAN used in experiment 5.

| Discriminator | Generator |
|---|---|
| Input: $28 \times 28$ grayscale image | Input noise vector $\boldsymbol{z} \sim \text{Unif}[0,1]^{62}$ |
| $4 \times 4$ conv. 64 LRELU. Stride 2. | FC. 1024 RELU. Batch norm. |
| $4 \times 4$ conv. 128 LRELU. Stride 2. Batch norm. | FC. $7 \times 7 \times 128$ RELU. Batch norm. |
| FC. 104 Leaky RELU. Batch norm. | $4 \times 4$ upconv. 64 RELU. Stride 2. Batch norm. |
| FC | $4 \times 4$ upconv. 1 channel. |

conv. refers to a convolution layer, FC means a fully-connected layer, RELU means a rectified linear unit, LRELU means Leaky RELU, and upconv is the transposed convolution.

*Remark 1.* The assumption that $P, Q$, and $R$ are all distinct in Theorem 1 is necessary for $\widehat{U_P^2}$ and $\widehat{U_Q^2}$ to follow a non-degenerate normal distribution asymptotically. If $R \in \{P, Q\}$, then $\widehat{U_S^2}$ for $S \in \{P, Q\}$ asymptotically follows a weighted sum of chi-squared random variables, and $U_S^2 = 0$. If $P = Q$, the covariance matrix in (1) is rank-defficient.

# D  Details of Experiment 5: Examining GAN Training

**LSGAN Architecture**   We rely on Pytorch code[5] by Hyeonwoo Kang to train the LSGAN [Mao et al., 2017] model that we use in experiment 5. Network architectures of the generator and the discriminator follow the design used in Chen et al. [2016, Section C.1]. We reproduce here in Table 2 for ease of reference.

**Kernel Function**   The kernel $k$ is chosen to be a Gaussian kernel on features extracted from a convolutional neural network (CNN) classifier trained to classify the ten digits of MNIST. Specifically the kernel $k$ is $k(\boldsymbol{x}, \boldsymbol{y}) = \exp\left(-\frac{\|f(\boldsymbol{x}) - f(\boldsymbol{y})\|_2^2}{2\nu^2}\right)$, where $f$ is the output (in $\mathbb{R}^{10}$) of the last fully-connected layer of a trained CNN classifier.[6] The architecture of the CNN is

$$\text{Input: } 28 \times 28 \text{ grayscale image} \rightarrow 5 \times 5 \text{ conv. 10 filters. } 2 \times 2 \text{ max pool}$$
$$\rightarrow 5 \times 5 \text{ conv. 20 filters. } 2 \times 2 \text{ max pool}$$
$$\rightarrow \text{FC. 50 RELU.}$$
$$\rightarrow \text{FC. 10 outputs.}$$

We train the CNN for 30 epochs and achieve higher than 99% accuracy on MNIST's test set. The Gaussian bandwidth $\nu$ is set with the median heuristic.

**Class Proportion of Generated Digits**   To examine the proportion of digits in the generated samples, we sample 4000 images from both models $P$ (LSGAN-15, LSGAN model trained for 15 epochs), and $Q$ (LSGAN-17, LSGAN model trained for 17 epochs), and use the CNN classifier to assign a label to each image. The proportions of digits are shown in Figure 7. We observe that the generated digits from both LSGAN-15 and LSGAN-17 follow the right distribution i.e., uniform distribution, up to variability due to noise. There is no mode collapse problem. This observation means that the difference between $P$ and $Q$ studied in experiment 5 in the main text is not due to the mismatch of class proportions.

Figure 7: Proportions of generated digits from the LSGAN models at 15th and 17th epochs. Classification of each generated image is done by a trained convolutional neural network classifier (see Section D).

(a) Samples from LSGAN trained for 15 epochs.

(b) Samples from LSGAN trained for 17 epochs.

Figure 8: Samples from LSGAN models trained on MNIST. Samples are taken from the models at two different time points: after 15 epochs, and after 17 epochs of training.

# E Comparing Different GAN Models Trained on MNIST

This section extends experiment 5 in the main text to compare other GAN variants trained on MNIST. All the GAN variants that we consider have the same network architecture as described in Table 2. We use the notation *AAA-n* to refer to a GAN model of type AAA trained for $n$ epochs. We note that the result presented here for each GAN variant does not represent its best achievable result.

**WGAN-GP-10 vs LSGAN-10**   Here we compare $P$ = Wasserstein GAN with Gradient Penalty [Gulrajani et al., 2017] and $Q$ = LSGAN [Mao et al., 2017] trained for ten epochs on MNIST. The results are shown in Figure 9. From the generated samples from the two models, it appears that LSGAN yields more realistic images of handwritten digits, after training for ten epochs. The positive power criterion values in Figure 9c further confirm this observation i.e., $Q$ is better at all digits.

(a) Sample from $P$ = WGAN-GP trained for 10 epochs.　(b) Sample from $Q$ = LSGAN trained for 10 epochs.　(c) Power criterion

Figure 9: Comparing WGAN-GP (Wasserstein GAN with Gradient Penalty) and LSGAN, trained for ten epochs on MNIST.

**GAN-40 vs LSGAN-40**   In this part, we compare $P$ = GAN-40 [Goodfellow et al., 2014] and $Q$ = LSGAN trained for 40 epochs on MNIST. The results are shown in Figure 10. It can be seen from visual inspection that LSGAN-40 is slightly better overall, except for digits 1 and 5 at which LSGAN-40 appears to be significantly better. This observation is also hinted by the power criterion values at digits 1 and 5 which tend to be positive (see Figure 10c).

(a) Sample from $P$ = GAN trained for 40 epochs.　(b) Sample from $Q$ = LSGAN trained for 40 epochs.　(c) Power criterion

Figure 10: Comparing GAN (the original formulation) and LSGAN, trained for 40 epochs on MNIST.

**WGAN-30 vs WGAN-30**   As a sanity check, we also run the same procedure on a case where $P = Q$. We set $P = Q$ = Wasserstein GAN (WGAN, [Arjovsky et al., 2017] trained for 30 epochs on MNIST. The results are shown in Figure 11. As expected, the power criterion values spread around zero in all cases. We note that we did not modify the procedure to treat this special case. In particular, in each trial, two samples are drawn from $P$ and $Q$ as usual.

(a) Sample from $P = Q = $ WGAN trained for 30 epochs.

(b) Power criterion

Figure 11: Comparing two models which are the same for sanity checking. The model is set to WGAN trained for 30 epochs.

## Footnotes

[4]Code for FSSD released by the authors: https://github.com/wittawatj/kernel-gof.

[5]https://github.com/znxlwm/pytorch-generative-model-collections (commit: 0d183bb5ea)

[6]Code to train the CNN classifier is taken from https://github.com/pytorch/examples/blob/master/mnist/main.py (commit: 75e7c75).