[Reviews · NeurIPS 2018]

Reviewer 1



The authors propose two statistical tests that can be used to testing goodness-of-fit for samples coming from two distributions. The proposed tests are combinations (and some generalization) of the work in Jitkrittum 16, 17b. The tests are then evaluated on toy datasets for their power and rejection rates, followed by their application to comparing different GANs. There is merit in the proposed models however the paper in its current form needs work. Presentation: Firstly much of the content in the section 2, specifically 88-104 and 110-130 needs to be wrapped up as prepositions or corollaries. It is confusing to follow through the text in its current form to arrive at the expressions for UME and FSSD, and the implicit MMD. Second, what is the summary of Theorem 1 and Theorem 2? To a statistical audience, the theorems themselves would ring bells, however, nips is also computer science and applied machine learning. Hence, the technical results needs to be summarized in few remarks. Lastly, a lot of time has been spent with the toy datasets. It is understandable that the authors want to portray the power of the tests on 'interpretable' datasets. However, much of this can be moved to the supplement, specifically later half of page 6. Evaluations: Isn't Rel-UME and weaker test than Rel-FSSD by definition? If so, then why are the rejection rates higher for the Rel-UME in some cases in Figure 4? What is the reasoning for choosing toy datasets 2 and 3? In the sense, do we expect the two tests to show different behaviour? Why does the saturation happen in Figure 4c) as sample sizes increase? Isn't FID-Diff a good criteria already (the magnitudes in the last columns in Table 1)? Application: This is the main issue. The technical results and their generalization of UME and FSSD are good, however the authors motivated the problem keeping in mind the application of GANs' comparison. However, section 4 is not suggesting if in fact the proposed tests are good to be deployed -- for problems 3-5, the baselines KID and FID are doing as well, one pair of GANs (and one setting) is used. Improving the presentation/evaluations and showing evidence on few GANs' comparison would surely boost the paper.

Reviewer 2



I have read the author response, and in particular, the authors have clarified my confusion regarding Figure 2. While I maintain my overall evaluation, I do feel that the contributions of this paper would be strengthened if either more theoretical results are provided for the proposed tests, or if the experiment results could be more convincing (as noted by another reviewer). ---------- The paper proposes two nonparametric statistical tests of relative goodness-of-fit that run in near-linear time, based on the unnormalized mean embeddings (UME) statistic and the finite-set Stein discrepancy (FSSD). By optimizing test locations to maximize test power, the proposed tests also yield informative features for distinguishing regions of the data domain in which one model fits significantly better than another. Experiments are conducted on toy problems and in comparing GAN models. Overall, the paper is fairly clearly written and easy to follow. While nonparametric goodness-of-fit testing has received much attention recently, few works investigate the topic of relative goodness-of-fit testing, which could be important for model comparison in practice, since “all models are wrong, but some are useful.” In this sense, I feel that this work is a nice addition to the existing literature. On the other hand, in terms of novelty, most of the techniques utilized in this work were developed in existing works: the definition of the UME and FSSD statistics as well as the idea of optimizing test locations to maximize test power in order to obtain informative features and reduce the computational complexity to near-linear time were all pioneered by Chwialkowski et al. and Jitkrittum et al., and the procedure to adjust the test threshold is similar to that in Bounliphone et al. The proofs of Theorems 1 and 2 are also fairly standard. I feel that the contributions of this paper would be strengthened if more theoretical results could be provided for the proposed Rel-UME, REl-FSSD tests in addition to deriving the asymptotic null distribution, such as an analysis of (asymptotic) test power which has been done for both the original UME and FSSD tests as well as MMD. Regarding the experiments, they seem fairly extensive and well-executed, although the presentation of the results (in particular the figures and table) could be improved. In particular, I found the “Rejection rate” label on the y-axes in the figures confusing at times, and it would be clearer to explicitly specify “Type I” or “Type II” error depending on the circumstance. Figure 4(a) is not vey informative, and it would be more interesting to zoom into the smaller sample-size region. Figure 2 (b) and (c) seem to be have been flipped (at least according to the description in the text); I also don’t get why Figure 2(b) is dominated by cat images with only a few airplane images---shouldn’t it consist mostly of airplane images since these are which the distribution does not fit well? The results presented in Table 1 also indicates that the current GAN comparison task might be too easy and it is hard to distinguish the performance difference of the various methods; perhaps it would be more informative to mimic Experiment 1 and consider setting the distribution R to be a mix of S and N images with varying proportions as well.

Reviewer 3



The paper develops two statistical tests, Rel-UME and Rel-FSSD, to measure the relative goodness of fit of two models. As shown in numerical experiments, both methods can control the false rejection rates below the nominal level, as well as achieve large powers with relative low computational costs. Some comments are listed as follows. 1. As shown in Figure 4, increasing J can increase the false rejection rates as well as the test power. In practice, how can we choose a proper J? 2. I suggest the authors add one more section ‘Discussion’ to discuss the advantages or limitations of the proposed statistical tests.